# Analgesic Action of Acetaminophen via Kv7 Channels

**DOI:** 10.3390/ijms24010650

**Published:** 2022-12-30

**Authors:** Jan-Luca Stampf, Cosmin I. Ciotu, Stefan Heber, Stefan Boehm, Michael J. M. Fischer, Isabella Salzer

**Affiliations:** Center for Physiology and Pharmacology, Medical University of Vienna, Währingerstrasse 13A, 1090 Vienna, Austria

**Keywords:** APAP, paracetamol, potassium channel, pain, inflammation, peripheral analgesia

## Abstract

The mechanism of acetaminophen (APAP) analgesia is at least partially unknown. Previously, we showed that the APAP metabolite N-acetyl-p-benzoquinone imine (NAPQI) activated Kv7 channels in neurons in vitro, and this activation of Kv7 channels dampened neuronal firing. Here, the effect of the Kv7 channel blocker XE991 on APAP-induced analgesia was investigated in vivo. APAP had no effect on naive animals. Induction of inflammation with λ-carrageenan lowered mechanical and thermal thresholds. Systemic treatment with APAP reduced mechanical hyperalgesia, and co-application of XE991 reduced APAP’s analgesic effect on mechanical pain. In a second experiment, the analgesic effect of systemic APAP was not antagonized by intrathecal XE991 application. Analysis of liver samples revealed APAP and glutathione-coupled APAP indicative of metabolization. However, there were no relevant levels of these metabolites in cerebrospinal fluid, suggesting no relevant APAP metabolite formation in the CNS. In summary, the results support an analgesic action of APAP by activating Kv7 channels at a peripheral site through formation of the metabolite NAPQI.

## 1. Introduction

Acetaminophen (APAP) is the most widely used analgesic worldwide and is listed on the WHO’s List of Essential Medicines for the treatment of light to moderate pain [1]. However, its exact mechanism of analgesic action remains unclear. A recent review provides several potential mechanisms [2], many of which involve metabolites of APAP. In the body, APAP adducts are formed by glucuronidation (APAP-Gluc) and sulfation, comprising 80–90% of modifications [3,4]. Alternatively, about 5–15% APAP is oxidized to N-acetyl-p-benzoquinone imine (NAPQI) via cytochrome P450 enzymes CYP3A4, CYP2E1 and CYP1A2 [5,6]. NAPQI is a highly reactive molecule, which results in a short half-life and only local reach [7,8]. This reactive electrophilic addition to cysteine residues has been demonstrated as a mechanism of action, e.g., on TRPA1 [7]. Inactivation occurs also through non-enzymatic coupling to glutathione, resulting in acetaminophen glutathione (APAP-GSH) [9], which can be quantified as an index of NAPQI production. In case of an APAP overdose, glutathione levels are depleted and NAPQI causes liver damage, primarily by impairing the mitochondrial electron transport chain [10,11].Voltage-gated potassium channels can be modulated by oxidative substances [12]. Previously, we demonstrated that NAPQI, but not APAP, activates Kv7 channels in neurons in vitro [13]. Kv7 channels pass potassium currents, which hyperpolarize the membrane potential and reduce the rate of action potentials fired in response to depolarizing stimuli [14]. The activation of Kv7 channels has been shown to induce analgesia [15]. Further, Kv7 channel opener retigabine has been shown to attenuate inflammatory pain, and this can be inhibited by Kv7 channel blocker XE991 [16]. Therefore, local NAPQI generation and activation of Kv7 channels might contribute to the analgesic effect of APAP. Together with our prior results of NAPQI on Kv7 channels, this motivated to investigate whether Kv7 contributes to APAP-induced analgesia in vivo, and if so, at which site in the body. Inflammation causes many changes that affect drug action. In a clinical trial, APAP reduced peripheral inflammation compared to placebo in oral surgery [17]. In another trial, no differences were observed between APAP and ibuprofen treatment after third molar surgery, considering swelling and pain intensity [18]. Analgesics are frequently used in conditions which have an underlying inflammatory component, therefore this study investigated naive and also inflamed conditions. In an inflamed environment, both mechanical and thermal hyperalgesia are common clinical symptoms [19], and reversing hyperalgesia is important for the clinical effect of an analgesic. The main hypothesis was that the Kv7-channel inhibitor XE991 would dampen the anti-nociceptive effect of APAP in inflammatory heat- and mechanical hyperalgesia in rats. In case there is a Kv7 component in analgesia, the second aim of this study was to identify the main site of action.

## 2. Results

### 2.1. Motor Coordination

RotaRod tests were performed on day 0 (habituation) and day 1 (baseline measurements). APAP (300 mg/kg), XE991 (3 mg/kg) or APAP + XE991 (300 mg/kg + 3 mg/kg) were analyzed for potential effects on motor coordination (factorial ANCOVA; main effects of APAP (*p* = 0.12) or XE991 (*p* = 0.46) did not indicate a substance-specific motor impairment (Figure 1).

### 2.2. Naive Condition

No change in mechanical or thermal sensitivity at the paws was observed one hour after s.c. application of APAP (300 mg/kg) and XE991 (3 mg/kg) (Factorial ANCOVA, in the absence of significant interactions, the respective main effects were *p* = 0.27 and 0.9, Figure 2, contralateral side in Figure 3). APAP did not interact with XE991 (*p* > 0.20).

### 2.3. Inflamed State

An injection of 100 µL λ-carrageenan 2% *w*/*v* into the right hind paw caused a consistent and substantial local hyperalgesia due to inflammation (Figure 2B,D). The inflammation was visually apparent by a swollen right hind paw, limping, and reduced activity. Mechanical withdrawal force was reduced from 31.8 ± 0.5 g before to 11.1 ± 2.5 g at 2.5 h after λ-carrageenan injection (mean ± SEM, *p* < 0.001, paired *t*-test), thermal withdrawal latencies were reduced from 9.1 ± 0.3 s before to 3.6 ± 0.3 s after λ-carrageenan injection (*p* < 0.001, paired *t*-test, Figure 2). Mechanical and thermal thresholds in the contralateral hind paw remained unaffected (*p* = 0.17 and 0.32, paired *t*-tests, Figure 3). There was a recovery of the inflammatory hypersensitivity at 8 and at 28 h after λ-carrageenan injection (Figure 1B,D).

#### 2.3.1. Mechanical Pain Thresholds during Hyperalgesia

A multivariate ANCOVA showed that XE991 modified the effect of APAP regarding hyperalgesia (Pillai’s trace 0.17, F = 4.1, *p* = 0.025). A subsequent separate analysis for mechanical thresholds showed that 1 h after injection, XE991 antagonized the antinociceptive effect of APAP (interaction APAP*XE991 *p* = 0.029). Contrasts between groups showed that APAP alone increased the paw withdrawal force by 8.4 g (95% CI 5.0–11.9 g, *p* < 0.001) compared to control. XE991 had no effect on its own compared to control (*p* = 0.43). In the presence of XE991, APAP increased paw withdrawal force only by 2.9 g (−0.5–6.4 g, *p* = 0.097). Thus, XE991 caused a reduction of the effect of APAP by 5.5 g (0.6–10.4 g, *p* = 0.029, Figure 2C). This point estimate corresponds to a 65% reduction of APAP effects by XE991.

#### 2.3.2. Thermal Pain Thresholds during Hyperalgesia

In contrast, the subsequent separate analysis for thermal thresholds showed no evidence for inhibition of the antinociceptive effect of APAP by XE991 1 h after injection (Interaction APAP*XE991 *p* = 0.18). Compared to control, APAP alone did not increase the withdrawal threshold (estimated increase: 1.4 s, 95% CI −0.4–2.8, *p* = 0.057), whereas in the presence of XE991, APAP increased the latency by 2.7 s (1.3–4.1, *p* < 0.001, Figure 2E). Thus, the effect of XE991 on APAP was +1.3 s (−0.6–3.3, *p* = 0.18). As the effect of APAP was not statistically significantly affected by XE991, its main (i.e., overall) effect was estimated. APAP increased the thermal withdrawal threshold (main effect *p* < 0.001), while XE991 had no effect (main effect *p* = 0.90).

### 2.4. Site of Action

To explore whether the Kv7 channels mediating the anti-nociceptive effects of systemic APAP 300 mg/kg application are located at a spinal site, the effects of intrathecal (i.t.) applied XE991 (20 µL of 55 µM, instead of s.c.) on mechanical hyperalgesia was probed. An ANCOVA with the paw withdrawal force values at timepoints 0 and 2.5 h as covariates resulted in an estimated increase of paw withdrawal force due to XE991 at 3.5 h by 1.6 g (95% CI −4.7–7.8, F = 0.27, *p* = 0.61, Figure 4A,B). Thus, spinally applied XE991 did not alter the inflammation-induced mechanical hyperalgesia to a relevant extent, pointing to a different location of action.

### 2.5. CSF and Tissue Analysis

To indirectly address the NAPQI occurrence, 300 mg/kg APAP was injected s.c. into anaesthetized rats, which were sacrificed after 15, 30 or 60 min. This resulted in detectable levels of APAP 15, 30 and 60 min after injection in liver and CSF (Figure 4C,D). NAPQI is spontaneously and non-enzymatically converted to APAP-GSH, wherever NAPQI and glutathione are available. In the liver, APAP-GSH was well detected. Despite a higher APAP concentration, there was no APAP-GSH in the CSF. Together with a substantial presence of glutathione in the CSF [20], this argues against relevant NAPQI levels in the central nervous system upon systemic APAP exposure, at least with an intact blood–brain barrier. One hour after subcutaneous injection of 3 mg/kg XE991, a measurement of blood, CSF and spinal cord levels was attempted. Samples spiked with XE991 1 µM allowed a detection at expected molecular weight. No corresponding signal was obtained from the unspiked samples.

## 3. Discussion

The Kv7 blocker XE991 inhibited an APAP-induced mechanical antinociception in a well-established inflammatory pain model, indicating a contribution of Kv7 channels to the action of APAP. The results suggest that the responsible Kv7 channel population is not located in the spinal cord.

### 3.1. Naive Condition

In the present study, application of APAP did not increase mechanical or thermal withdrawal thresholds in trials without inflammation. Prior results were mixed. For example, APAP was shown to increase thermal pain thresholds in Hargreaves and tail immersion tests [21,22]. Additionally, in naive mice, von Frey mechanical thresholds as well as noxious heat thresholds (tail immersion) were elevated [23,24]. On the other hand, paw withdrawal thresholds were reported to be unaltered by orally applied APAP 200 mg/kg in naive mice [25], and this is supported by a systematic review [26]. Presented RotaRod results might allow to hypothesize, that there is a mild trend towards a decrease in motor coordination by APAP 300 mg/kg. The dosing was based on the literature; a critical review thereof shows a report without change in motor coordination for APAP 300 mg/kg [25], but also a report with a RotaRod performance falling short of expectations for APAP 300 mg/kg [27].

For native human skin, heterogeneous results have been obtained as well; some studies suggested that APAP elevated heat pain thresholds [28,29], whereas others failed to detect a change [30,31]. It is unclear which factors might underlie these opposing results of antinociceptive actions of APAP in naive mammals. Mechanical von Frey pain thresholds were shown to be genotype-specific in healthy human subjects [32], but it seems rather hypothetical to assume that genetic differences account for the differences in the reported APAP-induced antinociception.

### 3.2. Inflamed Condition

APAP is known to attenuate mechanical and thermal hyperalgesia, and this applies to different models of established inflammation [26]. We found that blockage of Kv7 channels reduced APAP-induced analgesia for mechanical pain thresholds in hyperalgesic rats. The inflammation-induced reduction of thermal thresholds was partly resolved by APAP, however, this was independent of whether XE991 was administered. There are several differences between mechanical and thermal sensitivity and sensitization. The two physical stimuli activate different receptors in different neurons, connecting at least partially to different types of second order neurons, and trigger different signaling cascades, which has been carefully reviewed [33]. E.g. NMDA receptor modulation altered thermal but not mechanical hyperalgesia [34,35]. In patients, mechanical hyperalgesia leads to more disease burden than thermal hyperalgesia [36], and mechanical hyperalgesia is the best predictor for the intensity of musculoskeletal pain [37]. Therefore, it is important that the APAP-induced antinociception regarding mechanical stimuli was Kv7-dependent. In addition to the well-described spinal mechanisms, potential peripheral mechanisms could be considered, NAPQI might desensitize peripheral signaling involved in thermal transduction.

### 3.3. XE991

In our study, XE991 alone had no effect on pain in the naive or inflamed condition. In combination with APAP it reduced the antihyperalgesic effect for mechanical pain. In an expression system as well as in dorsal root ganglion neurons, XE991 inhibits Kv7-mediated currents [38]. Further, it acts only on activated Kv7 channels [39], which applies to electrically evoked Kv7 channel activity as well as to activity enhanced by NAPQI [13]. In inflamed paws, Kv7 activation increased weight bearing which could be antagonized by XE991, but the latter had no effect on its own [40]. This fits the results obtained for APAP and its inhibition of mechanical pain by XE991 in the present study. In sensory neurons dissected from naive and diabetic rats, XE991 increased mechanical and thermal currents. However, in the behavioral experiments of that study, XE991 reached significance level for the increase of mechanical thresholds but not for thermal thresholds [41], which is in line with our effects. Considering the two last mentioned studies, discordance remains as to whether XE991 effects on Kv7 are contingent on preceding activation of the channels, and this has been discussed in detail in a topical review [42]. In our experiments, there was no XE991 effect without pharmacological Kv7 stimulation.

### 3.4. Site of Action

Kv7 channels are present and functional in dorsal root ganglion and spinal dorsal horn neurons [40]. Kv7 channel activation by the APAP metabolite NAPQI in these neurons was demonstrated in vitro [13]. However, whether these neurons are exposed to APAP or metabolites thereof was not addressed. An APAP antinociception in the brain is supported by many studies, e.g., involving functional magnetic resonance imaging [21,43], and this component was neither addressed nor is it challenged in the present study. Rather, in vivo experiments with systemic APAP exposure but different modes of XE991 administration were designed to address whether peripheral and spinal Kv7 channels were involved. The analgesic action of systemic APAP was reduced by systemic, but not i.t. Kv7 inhibition, suggesting that Kv7-mediated analgesia is peripherally mediated. The XE991 application in this study was 0.49 µg, using an established i.t. volume of 20 µL and a concentration of 55 µM, which is well above the Kv7 channel IC_50_ of 0.26 µM [40], even with substantial dilution. I.t. application of XE991 25 µg/kg in young Wistar rats was above the threshold of affecting walking speed [44]. Based on their effect size, it might be assumed that the average of 1.8 µg/kg used in the present study has no motor side effects. However, an interpretation towards antinociceptive action appears unjustified, when the threshold for motor side effect is clearly exceeded, e.g., for XE991 36 µg/kg [45]. Kv7 channel inhibition by XE991 is state-dependent, favoring inhibition of activated channels [39]. In the present study, behavioral threshold testing causes neuronal activity, and by that Kv7 channel activation, therefore state-dependence of XE991 action should not have limited its effectiveness.

### 3.5. Tissues Analysis

After s.c. APAP injection in rats, the concentration of APAP in the CSF was higher than in the liver for all investigated timepoints. The proportions of conversion of APAP to its metabolites was described as about 90% sulfate/glucuronide adducts, 2% renal excretion and the remaining 8% are oxidized to NAPQI [3,4]. The latter was measured by the non-enzymatic GSH addition, resulting in APAP-GSH. The observed APAP-GSH at 7% of the APAP levels fits the expectation based on the literature. The observation that no APAP metabolite was found in the CSF would suggest that the metabolization of APAP is at least largely restricted to the periphery. However, it should be mentioned that the NAPQI metabolite L-cysteinyl-S-acetaminophen has been detected centrally in a similar experimental approach [9]. Our results suggest that a central effect of APAP is not caused by NAPQI, and thus probably does not involve Kv7 [13]. The HPLC-UV-MS/MS results support the hypothesis of a peripheral Kv7 effect caused by NAPQI. 

Reflecting the time of behavioral tests, measurements of blood, CSF and spinal cord were attempted one hour after subcutaneous injection of 3 mg/kg XE991. For a 350 g rat, a hypothetical distribution in 238 mL body water (68%, [46]) would result in 11.7 µM. The chosen protocol failed to detect XE991 in both blood, CSF and spinal cord, despite a detection limit of 0.25 µM. This cannot be explained by plasma half-life, as the duration of action of XE991 was similar to that of linopirdine, allowing to assume a similar half-life of 0.5 h in rat plasma [47]. However, it seems likely that the lipophilic XE991 (XLogP3-AA of 4.9) has a high distribution volume, and the resulting lack of detection in both fluids after one hour did not allow further conclusions.

### 3.6. Limitations

The Kv7-dependent fraction of APAP analgesia was estimated to be 65%. It should be considered that this has a large confidence interval, which is compatible with both small and large effects. Indocyanine green has been used to label the site of the i.t. injection. Indocyanine green is not known to alter pain thresholds, but a lack of knowledge does not exclude such an effect.

### 3.7. Conclusion

Kv7 channels are involved in the antihyperalgesic effect of APAP, and this is probably a peripheral effect mediated by NAPQI. This suggests an analgesic potential for peripherally restricted agonists of members of the Kv7 channel family.

## 4. Materials and Methods

### 4.1. Chemicals and Solutions

NaCl, 0.1 M NaOH and indocyanine green were obtained from Carl Roth GmbH & Co KG (Karlsruhe, Germany). XE991 >98% and λ-carrageenan were obtained from Sigma-Aldrich (St. Louis, MO, USA). Acetaminophen (APAP) was obtained from MedChemExpress (Monmouth Junction, NJ, USA). Acetaminophen glutathione disodium salt was purchased from Toronto Research Chemicals (Toronto, ON, Canada).

### 4.2. Preparation of Substances/Mixtures for Behavioral Experiments

APAP was ground to a fine powder and dispersed in 0.9% saline. XE991 was crushed to a fine powder and dispersed in 0.9% saline. A λ-carrageenan 2% *w*/*v* solution was prepared by dissolving 200 µg λ-carrageenan in 10 mL 0.9% saline under stirring at 45 °C. 0.5 % *w*/*v* indocyanine green was produced by dissolving 5 mg indocyanine green in 1 mL 0.9% saline. A 55 µM solution of XE991 was produced by adding 55 µL of 1 mM XE991 in dimethylsulfoxid (DMSO) to 945 µL 0.5% *w*/*v* indocyanine green. A DMSO control solution was prepared by adding 55 µL of DMSO to 945 µL 0.5% *w*/*v* indocyanine green.

### 4.3. Animals

Ethical approval for the experiment was granted by the Federal Ministry for Education, Science and Research Republic of Austria (BMBWF GZ 66.009/2020-0.121.290). Male and female Sprague Dawley rats (1:1) were obtained at an age of 6 weeks from Janvier and acclimatized in an inverted 12 h day/12 h night cycle with water and food ad libitum. At the age of 8 weeks, the rats were enrolled in the experiments, starting 1 h after the beginning of the dark period. Experiments were conducted in 7 Lux red light illumination, quantified by a PCE-LED 20 Lux Meter (PCE Instruments, Meschede, Germany). After the experiment, rats were anesthetized by ketamine 170 mg/kg and xylazine 12 mg/kg and euthanized by an intracardiac injection of 0.5 mL sodium pentobarbital 300 mg/mL according to the guidelines of the animal housing facility.

### 4.4. Behavioral Testing

An inflammatory model involving APAP-induced antinociception was established to investigate the mechanism involved. The animals were evaluated in a naive state and then in an acute inflammatory pain model, induced by λ-carrageenan injection. Prior to pain assessments, animals were tested on a Rat RotaRod (Ugo Basile, Gemonio, Italy) with a linear speed gradient from 10–40 rpm in 120 s to assess motor coordination. For all behavioral measurements, three replicates were acquired for every time point and the average of these was used for further analysis. To assess mechanical sensitivity, a Dynamic Plantar Aesthesiometer (linear gradient; 0–50 g in 12 s, Ugo Basile) was used. Thermal sensitivity was assessed using a Hargreaves Apparatus (Ugo Basile) with constant radiant heat of 190 kW/cm^2^ and a 20 s cut-off. The two experimenters performed the same measurements (mechanical or thermal) throughout all behavioral experiments and were blinded regarding the injected substances until the end of the experiments.

### 4.5. Experiment 1—APAP s.c. and Systemic Kv7 Inhibition

On day 0, rats underwent three runs for the RotaRod test and were habituated to the testing environment for the Dynamic Plantar Aesthesiometer and the Hargreaves Apparatus for 30 min each. Rats were split into six groups, one animal per group was tested in a randomized block design. On day 1 rats underwent a baseline RotaRod measurement followed by baseline measurements of mechanical and thermal pain thresholds, respectively. In a randomized and double-blind manner one of the following substances/mixtures was injected subcutaneously (s.c.) in the nape of the neck: APAP (300 mg/kg), K_V_7 inhibitor XE991 (3 mg/kg), APAP + XE991 (300 mg/kg + 3 mg/kg) or saline (0.9%). APAP was dosed as in a previous study investigating the mechanism of action [9], which is a high dose considering veterinary recommendations [48,49]. XE991 was also dosed as previously used and effective [50]. All substances were prepared in 0.9% saline and given in a volume of 2.5 mL/kg. One hour after injection, rats underwent a RotaRod test, followed by measurements of mechanical and thermal pain thresholds. On day 2, rats first underwent baseline measurements of mechanical and thermal pain. Following this, short-term anesthesia was induced with 4% isoflurane and a local inflammation was generated in the right hind paw by a s.c. injection of 100 µL 2% *w*/*v* λ-carrageenan in 0.9% saline. After 2.5 h mechanical and thermal pain thresholds were measured. Immediately thereafter, rats received a s.c. injection of the same treatment as the prior day. Mechanical and thermal pain thresholds were subsequently measured at the 3.5, 8 and 28 h timepoints after λ-carrageenan injection. (Figure 2A).

### 4.6. Experiment 2—APAP s.c. and Spinal Kv7 Inhibition

Based on experiment 1, a second experiment was performed to address whether a spinal site of action is responsible for the antinociception by APAP. On day 0, rats were habituated in the Dynamic Plantar Aesthesiometer enclosures for 30 min. On day 1, baseline mechanical pain thresholds were recorded. As in experiment 1, a local inflammation was induced by injection of 100 µL 2% w/v λ-carrageenan into the right hind paw. After 2.5 h mechanical thresholds were recorded. Thereafter rats were anesthetized with 4% isoflurane and received an intrathecal (i.t.) injection of 20 µL either control (DMSO) or 55 µM XE991, both in 0.9% saline supplemented with indocyanine green 0.5% w/v to stain the injection site. For a cerebrospinal fluid volume of about 300 µL [51] a homogenous dilution in that volume would result in a dilution to 3.7 µM XE991, but a more limited dilution and redistribution to a more lipophilic compartment appears more likely. Immediately after i.t. administration of XE991 or vehicle, rats received a s.c. injection of APAP (300 mg/kg). Mechanical pain thresholds were recorded 1 h after i.t. injection. Rats were euthanized after the experiment as described above and results were only included when the i.t. application to the desired site could be verified by examining the spinal cord for the presence of the dye.

### 4.7. Pharmacokinetic Experiments

Rats used in experiment 1 were anesthetized by ketamine 170 mg/kg + xylazine 12 mg/kg one day later and received a s.c. injection of 300 mg/kg acetaminophen or 0.9% saline. After 15, 30 or 60 min rats were euthanized by an intracardiac injection of 0.5 mL sodium pentobarbital 300 mg/mL according to the guidelines of the housing facility, with n = 3 for each timepoint, except for 0.9% saline control, with n = 1 for each timepoint. After euthanasia, the liver was removed, placed in phosphate-buffered saline (PBS), and stored at −80 °C. Cerebrospinal fluid (CSF) was sampled by cisternal puncture using an insulin syringe and stored at −80 °C. For XE991 analyses, three rats received a s.c. injection of 3 mg/kg XE991. Rats were euthanized after 60 min. Liver and CSF samples were obtained as described above. Spinal cord was acquired by dissecting the spine. All samples were stored at −80 °C.

#### 4.7.1. CSF Sample Preparation

CSF samples were diluted with an equal volume of HPLC-MS-grade H_2_O/Methanol/Acetone (50/25/25) and left at 4 °C for 20 min to precipitate proteins. After centrifugation for 10 min at 13,400× *g* and 4 °C, 100 µL of supernatant was transferred to a HPLC vial with glass insert and placed in the autosampler for subsequent triplicate analysis via HPLC-UV-MS/MS.

#### 4.7.2. Liver and Spinal Cord Sample Preparation

Liver samples were thawed and circa 50 mg of the outer liver lobe was cut off with a scalpel. Each liver sample was weighed into a 2 mL microtube. Spinal cord was weighed into a 2 mL microtube. A steel bead was added to each microtube as well as 500 µL HPLC-MS grade H_2_O. Samples were homogenized (Bertin Instruments Precellys Evolution, Montigny-le-Bretonneux, France) using a protocol of 2 × 20 s shaking with a 30 s pause. Methanol 250 µL and acetone 250 µL were added, the sample vortexed for 10 s, and proteins left to precipitate at 4 °C for 20 min. A volume of 150 µL was transferred to a new reaction tube and centrifuged for 10 min at 4 °C and 13,400× *g*. From the supernatant, 120 µL was placed in a new reaction tube, left at 4 °C for 20 min, and centrifuged for 10 min at 4 °C and 13,400× *g*. From the supernatant 100 µL were transferred to a HPLC vial with glass insert for subsequent triplicate analysis via HPLC-UV-MS/MS.

#### 4.7.3. HPLC-UV-MS/MS Analysis

Samples were analyzed using a Dionex UltiMate 3000 HPLC connected to a Compact Quadrupole Time-of-Flight (QTOF) mass spectrometer (Bruker, Billerica, MA). The mobile phases A and B were 0.1% TFA and Acetonitrile/0.1% TFA (90/10), respectively. Samples were separated using a phenomenex C18 150 mm x 30 mm 2.6 micron 100 Å column. The autosampler was set to 7 °C and the column compartment to 30 °C. A 25 min separation method was employed at flow rate of 0.4 mL/min and the following gradients: 0% B from 0 to 3 min, 0 to 20% B from 3 to 10 min, 20% B to 60% B from 10 to 13 min, 60% B to 100% B from 13 to 18 min, 100% from 18 to 19 min, 100% B to 0% B from 19 to 21 min, 0 % B from 21 to 25 min. A 10 min wash step was employed between each run. The column outflow passed through a 4-channel variable wavelength detector with wavelengths set to 214 nm, 243 nm, 254 nm and 280 nm. The HPLC outlet connected to the QTOF mass spectrometer which was equipped with an electrospray ionization source (ESI). The following parameters were set in the mass spectrometer: End plate offset 500 V, capillary voltage 4500 V, Nebulizer pressure 3 bar, dry gas 10 l/min, dry temperature 250 °C. Compounds were identified using their specific retention time, their mass, their fragment masses and their peaks in the UV detector. A spectrum library of the standards was created beforehand to identify the eluted compounds. A volume of 20 µL of a freshly prepared 10 µM mixture of APAP and APAP-GSH was injected into the HPLC-UV-MS/MS system and used to evaluate the retention times of each metabolite as well as collect their mass and fragment mass spectra for the compound library. CSF sample calibration was performed by pooling CSF saline samples and splitting them into six equivolume probes. APAP was added to make concentrations of 50 µM, 100 µM, 200 µM, 300 µM, 400 µM and 500 µM. An identical volume of H_2_O/Methanol/Acetone (50/25/25%) was added and left at 4 °C for 20 min to precipitate proteins. After centrifugation for 10 min at 13,400× *g* and 4 °C, 100 µL of supernatant was transferred to a HPLC vial with glass insert and placed in the autosampler for subsequent injection. Liver sample calibration was performed by generating 1 µM, 2 µM, 5 µM, 10 µM, 15 µM and 20 µM mixtures of APAP and APAP-GSH in HPLC-MS-grade H2O. Standards were transferred to HPLC vials with glass inserts and analyzed via HPLC-UV.

### 4.8. Sample Size Calculation and Statistical Analysis

#### 4.8.1. Sample Size Calculation of Experiment 1—APAP s.c. and Systemic Kv7 Inhibition

The published antinociceptive APAP effect after the first hour was considered [49]. The *p*-level was adjusted to 0.025, as both mechanical and thermal changes were compared for the groups, considering a power of 80%. The measured variable for mechanical sensitivity is the paw withdrawal force (measured in grams). λ-carrageenan treatment was expected to decrease mechanical sensitivity thresholds. Based on published literature, we expected pain thresholds (g, mean ± SD) in λ-carrageenan treated animals to be 10 ± 1.7 g and 19 ± 5.0 g in λ-carrageenan treated animals which received APAP [52]. Due to the addition of XE991, we considered a 40% reduction of the APAP-induced effect on the threshold as a relevant change. Thus, the relevant effect size is 3.6 g. The measured variable for thermal sensitivity is the paw withdrawal latency in seconds. Based on published literature, we expected a shortened paw withdrawal latency in λ-carrageenan treated animals in response to a noxious heat stimulus. We expected a similar analgesic effect by APAP on thermal pain thresholds. However, the lack of prior data did not allow estimating the variance, therefore the sample size calculation relied on the reported mechanical thresholds. Due to the addition of XE991, we considered a 40% decrease of the APAP-induced increase in withdrawal latency in XE991 and APAP treated animals as a relevant change. The sample size calculation was done with a calculator for comparing two independent means (https://www.stat.ubc.ca/~rollin/stats/ssize/n2.html, accessed on 1 March 2017), that is APAP and APAP+XE991, which demonstrates the role of XE991-inhibited K_V_7 channels. With an 80% power to detect a difference in means between these two groups and an expected pooled SD of 3.1, this results in a sample size of 12 in each of the groups. This sample size was used for all of the four groups.

#### 4.8.2. Analysis of Experiment 1

At first, an APAP effect modification by XE991 under inflamed conditions was tested for the mechanical and thermal thresholds at once by applying a multivariate (i.e., with two outcome variables), multivariable (i.e., with multiple predictors including continuous covariates) model (MANCOVA). It contained the fixed binary between-subject factors APAP and XE991, coding whether the compounds were applied in the animal or not, as well as the interaction term, testing the primary hypothesis. To adjust for interindividual differences before the treatment, the pre-treatment values at 0 and 2.5 h were used as covariates, allowing to estimate group differences as if all animals had exactly the same thresholds at 0 and 2.5 h. The separate analyses for each outcome were planned to be only conducted if the multivariate interaction term was significant with *p* ≤ 0.05. For the data analysis separated for each outcome, two general linear models were employed. The baseline measurement on day 2 (0 h) and carrageenan-induced sensitivity (at 2.5 h) were used as covariates with factors APAP and XE991 (ANCOVA). An interaction between the two factors was tested to reject the null hypothesis of K_V_7 channels not being involved in the APAP-induced effect. Significant interaction terms were broken down using contrasts. The a priori specified group difference between APAP and APAP + XE991 was tested as a primary hypothesis. In case of a non-significant interaction term, it was omitted from the model and the main effects were interpreted.

#### 4.8.3. Sample Size Calculation of Experiment 2—APAP s.c. and Spinal Kv7 Inhibition

The site of action was considered an independent, exploratory hypothesis as a peripheral action and insufficient penetration through the blood–brain barrier might occur as well as no peripheral action with only sufficient intrathecal concentration reached by intrathecal dosing. Comparison of XE991 with control regarding both mechanical and thermal thresholds in an unpaired t-test was used for the original power calculation with a *p*-level of 0.025. This calculation was updated with the decision to only test for mechanical sensitivity based on the results of experiment 1. With an updated *p*-level of 0.05, an APAP-induced antinociception of 8.4 g, a standard deviation of 2.9 g determined in experiment 1 and a 40% reduction thereof as relevant effect size, the calculation indicated 12 animals for a power above 80%. Considering an attrition rate by missing the injection target of 33%, the resulting sample size was calculated to be 18. The experiment ended when 18 animals had been included per group. The results were only then unblinded. In the post hoc visual inspection the fraction of injections without staining the target was slightly higher than anticipated, namely 39% and 44%, resulting in a final n of 11 and 10, respectively. This led to a reduction of power from 80 to 74% to detect the relevant effect.

#### 4.8.4. Analysis of Experiment 2

A general linear model was used to test whether the effect of APAP can be reduced by i.t. application of XE991. It included a binary factor coding whether XE991 was applied and two metric covariates, namely the baseline as well as the λ-carrageenan-induced mechanical sensitivity in g at 2.5 h.

#### 4.8.5. Additional Analyses

Analogous models were used for the analyses of data regarding the contralateral, non-inflamed mechanical and heat sensitivities. Animal numbers were balanced for sex. Analysis did not show any differences by sex as a factor. Statistical analysis was performed using IBM SPSS statistics 27. *p*-values ≤ 0.05 were considered statistically significant. Approximate normal distribution of residuals was checked graphically. Graphs were generated using GraphPad Prism 9 and arranged in CorelDraw 22.

## Figures and Tables

**Figure 1 ijms-24-00650-f001:**
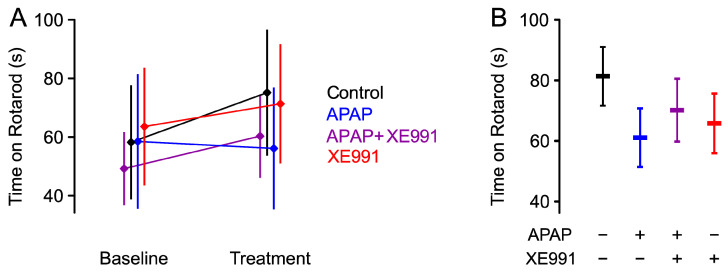
**APAP and XE991 do not impair motor coordination.** (**A**) Rats with systemically administered APAP (300 mg/kg), XE991 (3 mg/kg) or their combination, were tested for motor coordination on the RotaRod before (baseline) and one hour after dosing (treatment). Each data point is presented as the mean ± 95% CI of determinations in 12 rats. (**B**) Mean estimates ± 95% CI for time spent on the RotaRod 1 h after treatment, with baseline values as covariate.

**Figure 2 ijms-24-00650-f002:**
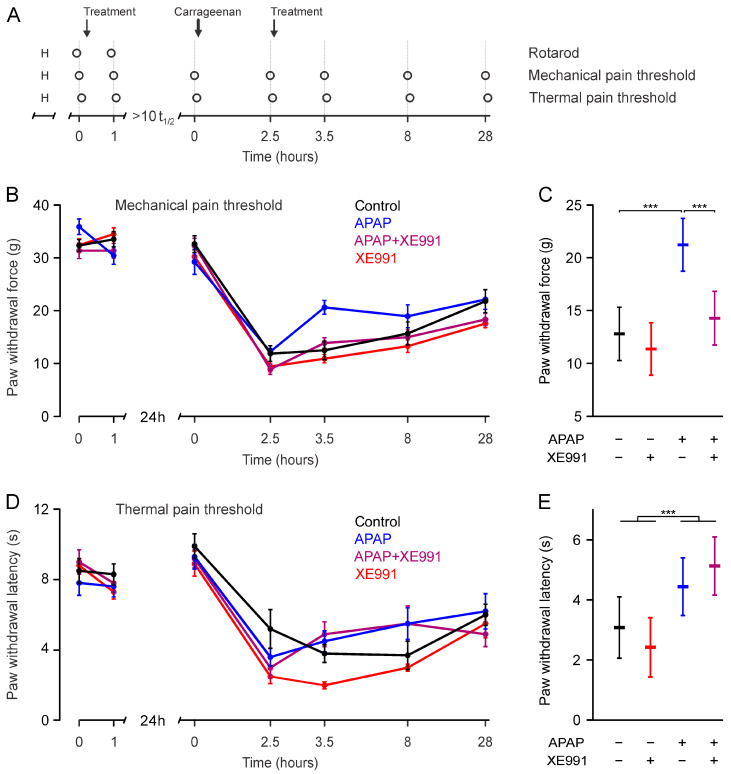
**APAP reduces mechanical hyperalgesia through Kv7 channels in rats.** (**A**) Timeline of the experiments. The circles indicate the time points of measurements. The arrows indicate the time points of s.c. substance injection and induction of hind paw inflammation by λ-carrageenan. The term treatment refers to either control (0.9% saline), APAP (300 mg/kg), APAP + XE991 (300 mg/kg + 3 mg/kg) or XE991 (3 mg/kg). All groups consist of 12 animals. “H” indicates the habituation on the day before the experiment. (**B**) Hind paw withdrawal latency in response to a linearly increasing mechanical stimulus at each timepoint for the indicated substances (mean ± SEM). (**C**) Mean and 95% confidence interval estimates of the withdrawal latencies for mechanical stimulation of the inflamed right hind paw 1 h after treatment (time point 3.5 h in panel **A**). *** *p* < 0.001 for the respective pairwise comparisons. (**D**) Thermal pain paw withdrawal latencies of the right hindpaw upon continuous infrared heating (Hargreaves test, mean ± SEM). (**E**) Estimated means and 95% confidence intervals for the withdrawal latencies for thermal stimulation of the inflamed right paw 1 h after treatment (time point 3.5 h in panel **A**). Measurements before λ-carrageenan (0 h same day) and 2.5 h after λ-carrageenan injection were used as covariates for estimation in panel (**C**,**E**). *** *p* < 0.001 for analgesic main effect of APAP, analyzed as there was no APAP*XE991 interaction (*p* = 0.18).

**Figure 3 ijms-24-00650-f003:**
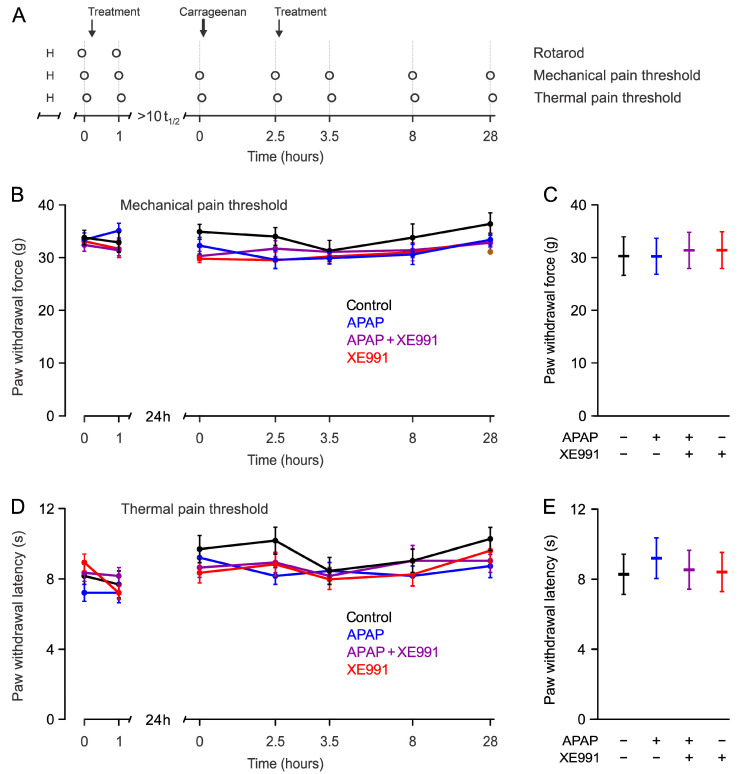
**No contralateral alteration of pain thresholds.** (**A**) Timeline of the experiments as above. The circles indicate the time points of measurements, contralateral paws were evaluated alternatingly with the λ-carrageenan-injected right paw. The arrows indicate the time points of s.c. substance injection as in Figure 2, namely control (0.9% saline), APAP (300 mg/kg), APAP + XE991 (300 mg/kg + 3 mg/kg), or XE991 (3 mg/kg). “H” indicates the habituation on the day before the experiment. (**B**) Hind paw withdrawal latency in response to a linearly increasing mechanical stimulus at each timepoint for the indicated substances. (**C**) Mean and 95% confidence interval estimates of the withdrawal latencies for mechanical stimulation of the inflamed right hind paw 1 h after treatment (time point 3.5 h in panel **A**). (**D**) Thermal pain paw withdrawal latencies of the right hand paw upon continuous infrared heating (Hargreaves test). (**E**) Mean and 95% confidence interval estimates for the withdrawal latencies for thermal stimulation of the inflamed right paw 1 h after treatment (time point 3.5 h in panel **A**). Measurements before λ-carrageenan (0 h same day) and 2.5 h after λ-carrageenan injection were used as covariates for estimation in panel **C** and panel **E**.

**Figure 4 ijms-24-00650-f004:**
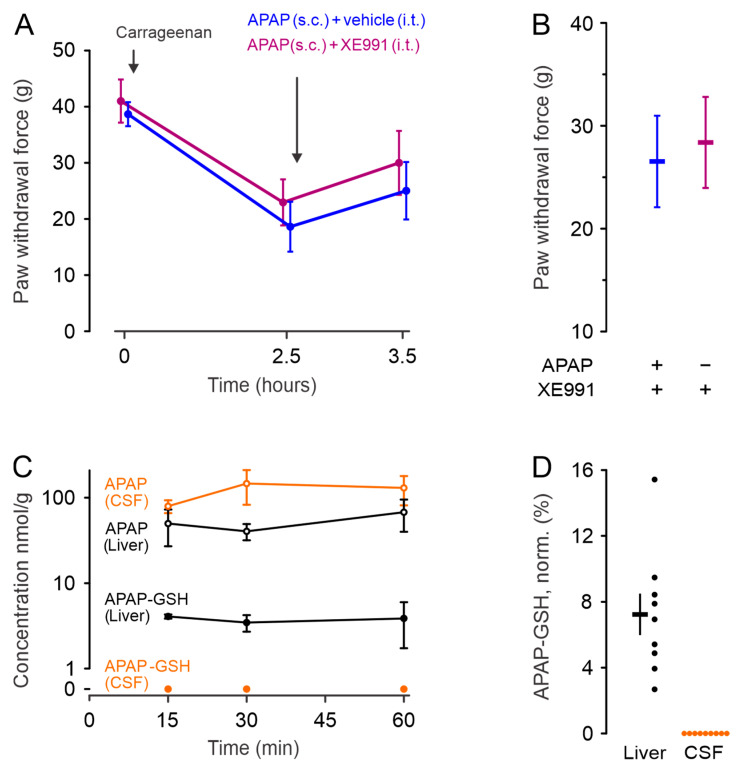
**Spinal Kv7 channels do not explain APAP analgesia in rats.** (**A**) As in the prior experiment, 2.5 h after λ-carrageenan injection mechanical withdrawal thresholds were lowered. APAP was injected s.c. (300 mg/kg). I.t. applied XE991 (20 µL of 55 µM) did not alter the inflammation-induced mechanical allodynia in comparison to the control (n = 11 and 10, respectively). Only animals in which the i.t. application could be verified after the experiment by the coinjected indocyanine green were considered. (**B**) Mean and 95% confidence interval estimates of the withdrawal force upon mechanical stimulation of the inflamed right hind paw 1 h after treatment (time point 3.5 h in panel **A**). (**C**) Time course of APAP and the non-enzymatic NAPQI-derivative APAP-GSH in liver and cerebrospinal fluid after s.c. injection of 300 mg/kg APAP. All data points are derived from three independent HPLC-UV-MS/MS measurements. (**D**) In the liver APAP-GSH levels were about 7% of the APAP levels. In contrast, no APAP-GSH was detected in the CSF. The panel shows the nine measurements from the three time points shown in panel **C**, normalized to the APAP levels found in this sample, as well as mean ± SEM. Glutathione levels in the CSF are about half of the plasma, but also half of the APAP-GSH would have been manyfold above the detection level. This indicates that systemic APAP does not generate a relevant NAPQI concentration in the CSF.

## Data Availability

The data presented in this study are available on request from the corresponding author.

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
