# Peer review of "Analgesic Action of Acetaminophen via Kv7 Channels"

_ijms, 2022, doi:10.3390/ijms24010650_

Round 1

Reviewer 1 Report

The article Analgesic action of acetaminophen via Kv7 channels Is accepted after consideration of the following minor comments.

1)      the full name of abbreviations should be mentioned where firstly appeared for example (line  38)

2)      abstract, in-vivo change to in-vivo.

3)      Abstract, authors should add conclusion.

4)      what is new in this study  even that reference 13 suppose authors proposal.

5)      The rational for authors hypothesis should be enhanced and supported by reported references.

6)      it is better to merge result and discussion.

7)      Line 147, what you mean by it

8)      Conclusion should improve and indicting the potential of this study.

Author Response

Reviewer 1

  1. the full name of abbreviations should be mentioned where firstly appeared for example (line 38)

The authors thank the reviewer for their thorough read of the manuscript. The full name of abbreviations have been mentioned where they first appear.

  1. abstract, in-vivo change to in-vivo.

The authors thank the reviewer for their thorough read, ’In vivo’ has been set in italics.

  1. Abstract, authors should add conclusion.

The authors thank the reviewer for their suggestion. We revised our conclusion sentence to make it clearer to “the results support an analgesic action of APAP by activating Kv7 channels at a peripheral site through formation of the metabolite NAPQI”.

  1. what is new in this study even that reference 13 suppose authors proposal.

The current study builds on Ray et al., which is explicitly mentioned in the manuscript (introduction line 49). This prior work contains only in vitro experiments. All in vivo approaches are new results. The study demonstrates a Kv7-dependence of the APAP analgesia in vivo. Further, the experiments point towards a peripheral effect.

  1. The rational for authors hypothesis should be enhanced and supported by reported references.

The rationale has been expanded by adding the effect of Kv7 channel opener retigabine “Further, Kv7 channel opener retigabine has been shown to attenuate inflammatory pain, and this can be inhibited by Kv7 channel blocker XE991” (Hayashi et al., Mol Pain. 2014; 10:15.)

  1. it is better to merge result and discussion.

The authors thank the reviewer for the suggestion. Without explicit editorial support for this change, the formatting guidelines indicate separate results and discussion sections.

  1. Line 147, what you mean by it

The authors thank the reviewer for the thorough reading. I.t. stands for “intrathecal”. The full name of the abbreviation has now been mentioned: “…the effects of intrathecal (i.t.) applied XE991 (20 µl of 55 µM, instead of s.c.) on mechanical hyperalgesia was probed.”

  1. Conclusion should improve and indicting the potential of this study.

The conclusion section was expanded by “This suggests an analgesic potential for peripherally restricted agonists of members of the Kv7 channel family.”

Reviewer 2 Report

In this manuscript, Stampf et al. investigated the mechanism of acetaminophen (APAP) analgesia. Base on their previous study showed APAP metabolite N-acetyl-p-benzoquinone imine (NAPQI) can activate neural KV7 channels ex vivo, they used APAP and KV7 channel blocker XE991 for in vivo experiments and discovered APAP can inhibit mechanical hyperalgesia and XE991 can blcok APAP’s analgesic effect on mechanical pain under the condition of λ-carrageenan-induced inflammation, while APAP has no effects on thermal pain under the same inflammation condition. They next discovered intrathecal XE991 cannot block the effects of APAP and there is no glutathione-coupled APAP in cerebrospinal fluid, which indicates that APAP analgesia may be a peripheral effect in the model used in the manuscript. The authors showed very interesting results in their animal model with comprehensive analysis. Here are some concerns.

1. The authors previously showed APAP metabolite NAPQI can activate Kv7 channels, and they aimed to study the APAP analgesia is through NAPQI’s effects on Kv7 channels. However, they did not use NAPQI in their in vivo assays. NAPQI should be an excellent positive control.

2. The authors detected APAP-GSH in the liver and cerebrospinal fluid (CSF) to indicate the NAPQI occurrence, and they didn’t detect the APAP-GSH in CSF, so they draw the conclusion “this argues against relevant NAPQI levels in the central nervous system upon systemic APAP exposure, at least with an intact blood 163 brain barrier”. However, like the authors showed in the scissionon part, in reference 9, they have detected the APAP-Cys in CSF to indicate the NAPQI occurrence, so whether APAP-GSH is not a good indicator for NAPQI?

Author Response

Reviewer 2

  1. The authors previously showed APAP metabolite NAPQI can activate Kv7 channels, and they aimed to study the APAP analgesia is through NAPQI’s effects on Kv7 channels. However, they did not use NAPQI in their in vivo assays. NAPQI should be an excellent positive control.

Indeed, NAPQI should be an excellent positive control. Unfortunately, due to its electrophilic nature, NAPQI is a highly reactive molecule. It modifies cysteine and other amino acid residues like tyrosine, tryptophan and methionine in vitro (Leeming et al., Chem Res Toxicol 2015; 28:2224–33) and reacts even with small molecules (Park et al., Antioxid Redox Signal 2013; 18:1713–22). Hence, we assumed that NAPQI injection s.c. or even intrathecally would form adducts in the vicinity of the injection site and therefore not reach the site to be investigated. In order to avoid this, we decided to block Kv7 channels using XE991, which has been established in in vivo experiments in former publications.

  1. The authors detected APAP-GSH in the liver and cerebrospinal fluid (CSF) to indicate the NAPQI occurrence, and they didn’t detect the APAP-GSH in CSF, so they draw the conclusion “this argues against relevant NAPQI levels in the central nervous system upon systemic APAP exposure, at least with an intact blood 163 brain barrier”. However, like the authors showed in the discussion part, in reference 9, they have detected the APAP-Cys in CSF to indicate the NAPQI occurrence, so whether APAP-GSH is not a good indicator for NAPQI?

We agree with the reviewer that APAP-Cys is a valid marker for NAPQI occurrence. However, extracted ion chromatograms and acquired MS/MS spectra did not reveal a presence of APAP-Cys or additional metabolite APAP-N-Acetylcysteine (m/z 271.1 ± 0.5 Da and m/z 313.0 ± 0.5 Da respectively) in any of our CSF samples.

There is evidence for a substantial presence of glutathione in the CSF (Kratzer et al., J Neurosci. 2018; 38:3466–79). Therefore, we decided to use APAP-GSH as marker for NAPQI presence. We acknowledge this discrepancy between the prior report and our results by “However, it should be mentioned that the NAPQI metabolite L-cysteinyl-S-acetaminophen has been detected centrally in a similar experimental approach”.

Round 2

Reviewer 2 Report

The authors have answered my questions, I feel the manuscript can be accepted in principle.